# Nonlinear Relationships among the Natural Environment, Health, and Sociodemographic Characteristics across US Counties

**DOI:** 10.3390/ijerph19116898

**Published:** 2022-06-04

**Authors:** Levi N. Bonnell, Benjamin Littenberg

**Affiliations:** Division of General Internal Medicine Research, Department of Medicine, University of Vermont College of Medicine, 89 Beaumont Ave S459, Burlington, VT 05405, USA; benjamin.littenberg@uvm.edu

**Keywords:** environment and public health, environmental epidemiology, mental health

## Abstract

**Background:** The aim of this study was to explore the nonlinear relationships between natural amenities and health at the intersection of sociodemographic characteristics among primary care patients with chronic conditions. **Methods:** We used survey data from 3409 adults across 119 US counties. PROMIS-29 mental and physical health summary scores were the primary outcomes. The natural environment (measured using the County USDA Natural Amenities Scale (NAS)) was the primary predictor. Piecewise spline regression models were used to explore the relationships between NAS and health at the intersection of sociodemographic factors. **Results:** We identified a nonlinear relationship between NAS and health. Low-income individuals had a negative association with health with each increase in NAS in high-amenity areas only. However, White individuals had a stronger association with health with each increase in NAS in low-amenity areas. **Conclusions:** In areas with low natural amenities, more amenities are associated with better physical and mental health, but only for advantaged populations. Meanwhile, for disadvantaged populations, an increase in amenities in high-amenity areas is associated with decreases in mental and physical health. Understanding how traditionally advantaged populations utilize the natural environment could provide insight into the mechanisms driving these disparities.

## 1. Introduction

There is a vast literature on the built environment and health [1]. Constructs, such as population and housing density [2], access to healthy food [3,4,5], proximity to walkable destinations and transit [6], and varied land use [7], can bolster active travel and a healthful diet [8], improving mental and physical health and obesity. The literature on the *natural* environment and health is less developed.

Similar to the built environment, the natural environment is a multifaceted construct. Traditionally, the natural environment has been interpreted in terms of toxicity, focusing on how air pollution [9], climate change [9], natural disasters [10], and agricultural chemicals [11] negatively impact health; in terms of beneficence, focusing on the healthful benefits of exposure to nature and, more recently, urban greenspace [12]; and in terms of the “biophilia hypothesis”, where humans possess an innate tendency to connect with nature [13,14]. Fewer studies focus on the nonmodifiable domains of the natural environment in which individuals reside, such as topography and climate. One study used weather station data and found that populations that reside in places with a better climate had a lower Body Mass Index (BMI), seemingly through increased physical activity [15]. Another study found an inverse association between county natural amenities and BMI using several large datasets [16]. Although these factors influencing health are nonmodifiable, if we can identify the differences between populations, there may be ways to reduce inequalities and improve health.

The relationships between the environment and health are complex and, therefore, may require complex modeling. Only recently have studies started to include models that allow for the possibility of nonlinear relationships. Three recent studies found convincing evidence that aspects of the built environment and walkability are not monotonically related to BMI. Each study found that increasing walkability was associated with increased BMI in areas with lower walkability, but an inflection point was reached, where increasing walkability was associated with decreased BMI in areas with higher walkability [17,18,19]. Another study found a similar nonlinear relationship with walkability and mental and physical health. To the best of our knowledge, no studies have assessed whether the relationship between the natural environment and health is nonlinear.

Many studies in this realm focus on children or the elderly because they are assumed to be more dependent on their local environment, but very few studies focus on adults with chronic conditions. It is unclear if the natural environment influences health differently for different populations.

We sought to explore the relationship between the natural environment and health at the intersection of various demographic factors and the social determinants of health among primary care patients with multiple chronic conditions, allowing for the possibility of a nonlinear relationship. Similar to previous built environment work, we hypothesized that there may be a nonlinear relationship between the natural environment and health and that these relationships may differ by certain patient characteristics. This was an exploratory analysis; there was no *a priori* hypothesis of the shape of the nonlinear relationship.

## 2. Methods

### 2.1. Data and Setting

We used pre-COVID-19, baseline survey results from a multi-center, randomized trial of primary care patients, described elsewhere [20]. Data were collected from 3929 adults with chronic conditions (heart disease, diabetes, lung disease, arthritis, mood disorder, insomnia, substance abuse, chronic pain, or irritable bowel syndrome) from 44 primary care practices across 13 states. Records were included in this sub-study if they provided a home address and had complete data for the primary predictor and outcomes. After exclusions, the final analytic sample had 3409 records from 119 US counties.

Our primary outcomes were mental and physical health summary scores measured using the PROMIS-29 [21,22], a validated survey that assesses emotional, physical, and social function, as well as well-being. The PROMIS-29 produces mental and physical health summary t-scores from 0 (poor health) to 100 (excellent health), which are standardized to the US population with a mean of 50 and a standard deviation of 10.

The primary predictor was the natural environment measured using the USDA Economic Research Service’s Natural Amenities Scale (NAS), a county-level composite score derived from winter and summer temperatures, winter sunlight hours, summer humidity, topographical variation, and total water area [23,24]. This scale ranges from −6.4 (Red Lake, MN, USA) to 11.2 (Ventura, CA, USA) overall, and from −2.4 (Lexington, KY, USA) to 9.8 (San Diego, CA, USA) in this sample. Higher values represent more attractive natural amenities. The top ten scoring counties are in California, while the ten lowest scoring counties are in Indiana, North Dakota, and Minnesota. Alaska and Hawaii are not included in NAS. NAS is an empirical construction that describes the revealed preferences of US adults and estimates retiree population change. Traditionally, in the United States, retirees tend to migrate toward places with warmer winters, mild summers, varied topography, and access to water features.

Clinical knowledge and the prior literature informed the selection of subgroups, including older age (<65 vs. ≥65 years), sex (male vs. female), race (White vs. other), ethnicity (Hispanic vs. non-Hispanic), marital status (currently married vs. not), low annual household income (<USD 30,000 vs. ≥USD 30,000), education (college graduate or more vs. associate degree or less), employment status (employed or retired vs. not), and rural/urban status (rural vs. urban as defined by Rural Urban Commuting Area (RUCA) codes) [25]. Race and ethnicity are considered different constructs in the United States to allow for the classification of individuals within any race and simultaneously as Hispanic or non-Hispanic cultural groups.

### 2.2. Geocoding

Each participant’s household was geocoded and assigned the corresponding NAS score. Latitude and longitude points were assigned for each participant’s home address using the ArcGIS address geocoder (ESRI Inc., Redlands, CA, USA) and the WGS 1984 coordinate system. Addresses that had less than a 100% match were manually checked for errors. After removing records without an address and manually geocoding, 100% were matched to a county. Rural routes and P.O. Boxes that could only be matched to a zip code centroid were included as zip code centroids nested within counties.

### 2.3. Statistical Analysis

Based on previous work [16,17], we hypothesized that the relationship between the natural environment and health may be nonlinear. However, there was no *a priori* shape in mind. We used locally weighted scatterplot smoothing (LOWESS) curves to make first-order estimations of the best fit [26]. The LOWESS function explores the relationship between two variables by fitting many simple models to various subsets of the data, resulting in a unique, nonlinear visual description of the relationship. The resulting graph identified a clear inflection point near NAS = 0 for both mental and physical health, indicating that a piecewise linear regression may be appropriate (see Figure 1). We confirmed nonlinearity by comparing linear regressions and piecewise linear spline regressions with a knot at zero using the Akaike Information Criterion (AIC) [27]. Once confirming the best-fit model, we investigated different subgroups using the same modeling technique. Regression models have added advantages over LOWESS in interpretability and statistical testing. We forced the model to have continuity at the inflection point. After fitting unadjusted models, we performed multivariable analyses controlling for age, sex, race, ethnicity, marital status, income, education, employment status, and rural/urban status. Subgroups were investigated if the interaction term had a *p* < 0.20 (this was not a statistical test, simply a cutoff). For the subgroup models, the moderating term was omitted as a covariate in the model. There were 119 counties represented in this study, but the majority (90%) of participants resided in 24 counties. Therefore, we performed a sensitivity analysis using only participants residing in the 24 counties.

All tests were two-tailed, and the threshold for statistical significance was set at α = 0.05. Stata 16.1 (StataCorp LP, College Station, Texas) was used for data management and statistical analysis.

All study procedures were approved by The University of Vermont Committees on Human Research (CHRMS#16-554). Informed consent was provided by all study participants.

## 3. Results

The participants’ mean age was 64 years old, and 46% of participants were aged <65. The sample was primarily female (63%), non-Hispanic (92%), White (79%), and unemployed or retired (67%). About half of the participants were unmarried (51%), had a low income (50%), and did not graduate college (53%). The mean physical and mental health summary scores were 46 and 50, respectively, while the average US population average is 50. The mean NAS score was 1.8, while 33% of participants lived in low-amenity areas (NAS < 0). Many participant characteristics differed between low and high amenities (see Table 1).

The piecewise linear spline models produced lower AIC values for physical (AIC = 25,083) and mental (AIC = 24,558) health than the linear models did (AIC = 25,110 and AIC = 24,569, respectively), suggesting that the nonlinear models had a lower prediction error and a better fit. Upon visual inspection, the piecewise linear spline model closely approximated the LOWESS curve (see Figure 1 and Appendix A). For both mental and physical health, we found a “hockey stick”-shaped curve. In areas with lower natural amenities, an increase in amenities was associated with better health, but in higher amenity areas, health did not change with additional amenities. Specifically, in low-amenity areas (NAS < 0), more amenities were associated with better physical (ß = 1.76, 95% confidence interval (CI) 1.17, 2.36) and mental (ß = 1.08, 95% CI 0.53, 1.63) health. However, in high-amenity areas (NAS ≥ 0), more amenities were not associated with physical (ß = −0.01, 95% CI −0.09, 0.12) or mental (ß = −0.00, 95% CI −0.09, 0.09) health (see Table 2).

Unadjusted analyses were also performed for subgroups. Based on the significance of interaction terms, income, race, ethnicity, and rural/urban status subgroups were investigated for both mental and physical health; marital status was assessed for physical health; and education was assessed for mental health. In high-amenity areas only, low-income individuals had a negative association between amenities and mental health with each additional NAS. Furthermore, living in rural high-amenity areas was associated with lower mental health, while living in urban high-amenity areas was associated with lower physical health. In low-amenity areas only, non-Hispanic White individuals had a stronger association between amenities and mental and physical health than did non-White and Hispanic individuals (see Figure 2 and Appendix A). Those living in urban low-amenity areas had an improvement in mental health, while those living in rural low-amenity areas had an improvement in physical health for each increase in NAS. Furthermore, in low-amenity areas only, married individuals had an improvement in physical health, and those with a higher education had an improvement in mental health compared to their counterparts (see Table 2).

After adjusting for relevant confounding variables, the overall relationship between NAS and health was attenuated and no longer significant. However, there were still important and consistent associations within the subgroups. Similar to the unadjusted analysis, in low-amenity areas only, White individuals had a stronger association between amenities and mental and physical health than non-White individuals. However, in high-amenity areas, White individuals had a negative association between amenities and mental and physical health for each additional increase in natural amenities, and this was especially prominent among low-income individuals. Furthermore, living in a rural high-amenity area was associated with worse physical health for each additional increase in NAS. The slopes differed significantly between low- and high-amenity areas for many of the models (see Table 3).

We performed a sensitivity analysis using 90% of the participants that resided in 24 counties. No results significantly changed. For instance, in the unadjusted analysis, in low-amenity areas, the coefficient was from 1.76 to 1.80 for physical health and from 1.08 to 1.09 for mental health. Similar small differences were observed in high-amenity areas. In the adjusted analysis, in high-amenity areas, the coefficient for mental health became slightly more negative but lost significance, likely due to the decrease in sample size. No other notable changes were noticed in the sensitivity analysis.

## 4. Discussion

Previous research found nonlinear relationships between the *built* environment and health. Here, we extend these findings to the *natural* environment and to various subgroups. In this exploratory analysis, we found that the effect of the natural environment on physical and mental health was important in areas with lower natural amenities but not in areas with higher natural amenities. After adjustment, these relationships appeared to be significant, especially among more advantaged populations. In low-amenity areas, White individuals seemed to benefit from an improving natural environment, while in higher amenity areas, the health of lower income individuals *decreased* with improving amenities.

The relationship between the natural environment and health is complex, and there are likely several mechanisms at work. In general, as seen here, health improves as the natural environment improves. A better climate and more varied topography can lead to more physical activity and better health [15,16]. Residing near green space can increase physical activity and subsequently improve health [28]. Simply being “in nature” can reduce stress and lead to lower blood pressure, anxiety, depression, and risk of poor health outcomes [29,30,31]. In turn, residing in places with high air pollution or near large agricultural farms can lead to health issues [32,33]. There is also a budding literature on geographical psychology focusing on how individual psychological characteristics interact with the local environment and ultimately affect health [34,35,36]. A possible explanation for the nonlinear association in our study is that once some criteria are met for a favorable natural environment (milder, sunnier winters; less humid summers; some variation in topography; or access to some blue and green space), other county-level competing factors (urban/rural continuum, built environment, and cost of living) come into play. However, why, then, do advantaged populations have health benefits in low-amenity areas while disadvantaged populations do not? Perhaps advantaged populations in low-amenity areas can focus on health and physical activity through means outside the natural environment (gyms, yoga, etc.). Meanwhile, in high-amenity areas, low-income individuals have a decline in mental and physical health with additional amenities, while there is no impact on higher income individuals. This could be due to increases in the cost of living associated with high-amenity areas. Low-income individuals may need to work multiple jobs to make ends meet and, therefore, have less time or money to take advantage of the natural environment.

As expected in a population with chronic conditions, the average physical health was lower than that of the average US population. The mental health scores were similar to those of the US population, even though we included individuals with known behavioral conditions. In the worst natural environments, however, physical and mental health were much worse than those at the national level. Improving the natural environment in which one resides from −4 to 0 on NAS is associated with an increase in mental health and physical health, although only for White individuals after adjustment.

There are important policy implications if this work can be confirmed. Although most natural amenities are nonmodifiable, there are strategies to improve health by reducing the health benefit inequities between certain groups that are associated with improvements in the natural environment. One could improve some aspects of the natural environment by adding green space, recreation areas, and hiking trails, which have been shown to increase physical activity and improve health [18]. It will be important to identify potential mechanisms that drive improved health in low-amenity areas for advantaged populations. Because we only see improvements in health for traditionally advantaged populations, perhaps identifying how these populations are benefiting could inform outreach campaigns geared toward disadvantaged populations.

There are several limitations to this study. The spatial granularity of NAS is low and may not accurately measure the natural environment at the sub-county level. However, many of the characteristics that make up NAS probably do not differ drastically within counties. The theory behind the creation of NAS may not generalize outside the United States, as other counties may find colder winters appealing. Furthermore, there is a discordance in dates between the survey data (2018) and NAS (2000). Although survey data are self-reported, the PROMIS-29 is validated and reliable. We collected data from 44 primary care practices of various sizes, structures, and settings across the United States. However, not all states or regions are represented, and, therefore, the results of this study may not be generalizable to areas outside the study area. Most of the range of the natural amenity scale is represented. Although 119 counties are represented in this study, the majority (90%) of participants resided in 24 counties, further hampering generalizability and questioning whether the findings of this study are an artifact of the data. However, a similar inflection point has been shown using a nationwide dataset [16]. Future research should investigate why the inflection point is at zero and explore more complex modeling techniques, such as general additive models (GAMs). Furthermore, all study participants have multiple chronic conditions, which limits generalizability to otherwise healthy adults. As with most studies of the environment, these data cannot distinguish if the environment affects health or if health and sociodemographic characteristics influence where people live.

There are many strengths to this study. While most studies in this realm focus on children or the elderly, we examined a large sample of adult primary care patients with chronic conditions. While other studies have relied on aggregated health information that may suffer from the ecologic fallacy [37], we used individual-level data. This may be the first study to investigate nonlinear relationships between the natural environment and health. Future studies should include information on the cost of living and explore how built, natural, and social environments affect health.

In this nationwide analysis of adults with chronic conditions, we found that the natural environment affects health differently depending on the number of natural amenities available. Furthermore, the benefits of the natural environment are not homogenous across different populations. Understanding why these differences exist could lead to strategies to improve health through improving equitable access to the natural environment and, ultimately, to improved mental and physical health.

## Figures and Tables

**Figure 1 ijerph-19-06898-f001:**
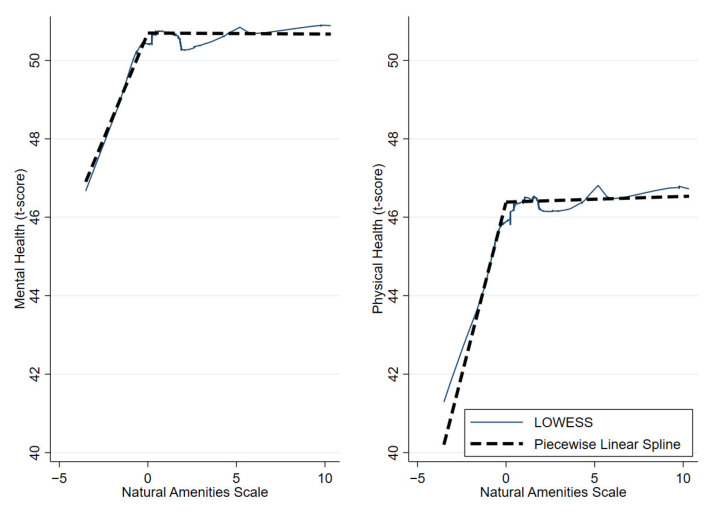
LOWESS curves and piecewise linear splines visualizing the unadjusted relationships among NAS and mental and physical health.

**Figure 2 ijerph-19-06898-f002:**
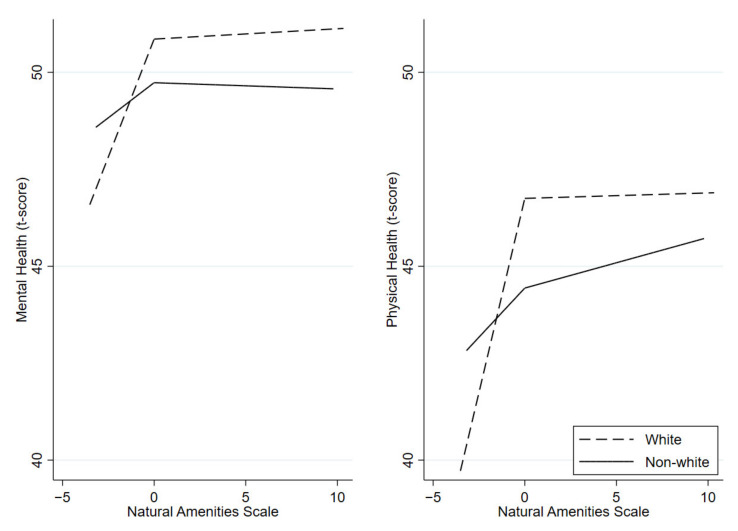
Unadjusted piecewise linear spline models visualizing the relationships between NAS and mental and physical health for White (dashed lines) and non-White (solid lines) individuals.

**Table 1 ijerph-19-06898-t001:** Participant characteristics stratified by the natural amenities scale.

	Low Amenities NAS < 0	High Amenities NAS ≥ 0	*p*
N	1140	2269	
Mean age ±SD	64 ± 12	63 ± 14	0.12
Sex, female	734 (65%)	1398 (62%)	0.11
Race, White	769 (69%)	1873 (84%)	<0.001
Ethnicity, Hispanic	57 (5%)	204 (9%)	<0.001
Marital status, married	515 (45%)	1148 (51%)	0.003
Employment, working	372 (33%)	767 (34%)	0.47
Income, <USD 30 k/year	666 (58%)	1055 (47%)	<0.001
Education, college graduate or more	429 (38%)	1183 (52%)	<0.001
Mean physical health summary score ± SD	45 ± 10	46 ± 10	<0.001
Mean mental health summary score ± SD	50 ± 9	50 ± 9	0.96

**Table 2 ijerph-19-06898-t002:** Unadjusted piecewise spline regression overall and subgroup models.

	Mental Health ß (95% CI)		Physical Health ß (95% CI)	
	Low Amenities (NAS < 0) ß (95% CI)	High Amenities (NAS ≥ 0) ß (95% CI)	Low Amenities (NAS < 0) ß (95% CI)	High Amenities (NAS ≥ 0) ß (95% CI)
	Unadjusted models			
Simple model	**1.08 (0.53, 1.63) ***	−0.00 (−0.09, 0.09) *	**1.76 (1.17, 2.36) ***	−0.01 (−0.09, 0.01) *
Subgroups				
Low income, y	0.13 (−0.50, 0.76) *	**−0.17 (−0.33, −0.02) ***	0.50 (−0.14, 1.12)	−0.14 (−0.29, 0.01)
Low income, n	0.40 (−0.70, 1.51)	−0.00 (−0.11, 0.11)	1.15 (−0.07, 2.41)	−0.02 (−0.15, 0.11)
White race, y	**1.21 (0.51, 1.91) ***	0.02 (−0.08, 0.13) *	**2.00 (1.23, 2.77) ***	0.01 (−0.11, 0.14) *
White race, n	0.36 (−0.67, 1.39)	−0.02 (−0.23, 0.20)	0.51 (−0.54, 1.56)	0.13 (−0.09, 0.35)
Hispanic, y	1.13 (−1.27, 3.52)	−0.03 (−0.36, 0.30)	1.40 (−1.07, 3.87)	−0.03 (−0.37, 0.31)
Hispanic, n	**1.06 (0.48, 1.68) ***	0.01 (−0.36, 0.30) *	**1.75 (1.13, 2.37) ***	0.02 (−0.09, 0.13) *
Married, y	--	--	1.51 (0.40, 2.63) *	0.00 (−0.15, 0.16) *
Married, n	--	--	**1.07 (0.26, 1.88)**	0.06 (−0.10, 0.21)
Graduated college, y	**1.63 (0.61, 2.65) ***	−0.05 (−0.17, 0.06) *	**--**	--
Graduated college, n	0.56 (−0.11, 1.23)	−0.16 (−0.33, 0.01)	**--**	--
Rural residence, y	1.59 (−0.22, 3.40)	**−0.83 (−1.51, −0.16)**	**1.97 (1.34, 2.61)**	0.06 (−0.04, 0.17)
Rural residence, n	**1.09 (0.51, 1.67)**	0.03 (−0.07, 0.13)	1.53 (−0.39, 3.45)	**−1.73 (−2.45, −1.02)**

Each coefficient (ß) presented is the linear slope of health as a function of NAS across a range of amenities. For instance, mental health was positively associated with NAS in low-amenity areas with a slope of 1.08 but not with high-amenity areas where the slope was 0.00. Slopes that significantly differ from zero are shown in bold type. * Significant difference (*p* < 0.05) in slopes between low- and high-amenity areas.

**Table 3 ijerph-19-06898-t003:** Multivariable piecewise spline regression overall and subgroup models.

	Mental Health ß (95% CI)		Physical Health ß (95% CI)	
	Low Amenities (NAS < 0) ß (95% CI)	High Amenities (NAS ≥ 0) ß (95% CI)	Low Amenities (NAS < 0) ß (95% CI)	High Amenities (NAS ≥ 0) ß (95% CI)
	Adjusted models			
Full model	0.30 (−0.28, 0.88)	**−0.09 (−0.20, 0.00)**	0.50 (−0.12, 1.13)	−0.05 (−0.15, 0.06)
Subgroups				
Low income, y	0.64 (−0.08, 1.36) *****	**−0.27 (−0.44, −0.10) ***	0.64 (−0.09, 1.37) *	**−0.21 (−0.39, −0.04) ***
Low income, n	−0.07 (−1.18, 1.04)	−0.01 (−0.11, 0.13)	0.43 (−0.83, 1.69)	0.04 (−0.10, 0.17)
White race, y	**0.75 (0.04, 1.46)** *	−0.08 (−0.19, 0.02) *	**1.03 (0.25, 1.81) ***	−0.07 (−0.19, 0.05) *
White race, n	0.00 (−1.14, 1.15)	−0.11 (−0.35, 0.13)	−0.13 (−1.26, 1.01)	0.00 (−0.23, 0.24)
Hispanic, y	0.01 (−2.81, 2.83)	0.00 (−0.35, 0.36)	0.31 (−2.52, 3.13)	−0.07 (−0.42, 0.29)
Hispanic, n	0.45 (−0.16, 1.05)	**−0.10 (−0.21, −0.00)**	0.57 (−0.09, 1.22)	−0.06 (−0.17, 0.06)
Married, y	--	--	0.62 (−0.50, 1.73)	−0.05 (−0.20, 0.10)
Married, n	--	--	0.45 (−0.32, 1.22)	−0.05 (−0.21, 0.10)
Graduated college, y	0.40 (−0.64, 1.45)	−0.09 (−0.21, 0.02)	--	--
Graduated college, n	0.36 (−0.37, 1.10)	−0.08 (−0.26, 0.10)	--	--
Rural residence, y	0.62 (−1.15, 2.39)	0.13 (−0.67, 0.70)	0.51 (−1.38, 2.40)	**−0.84 (−1.57, −0.10)**
Rural residence, n	0.30 (−0.34, 0.94)	−0.07 (−0.18, 0.03)	0.64 (−0.04, 1.33)	−0.02 (−0.13, 0.09)

Models were adjusted for age, sex, race, ethnicity, marital status, education, and employment. In the subgroup analysis, moderating variables were omitted. Slopes that significantly differ from zero are shown in bold type. * Significant difference (*p* < 0.05) in slopes between low- and high-amenity areas. Regression coefficient ß representing the linear relationship between NAS and health.

## Data Availability

The data presented in this study are available on request from the corresponding author. The data are not publicly available yet because data are still being cleaned.

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
