# Peer review of "Nonlinear Relationships among the Natural Environment, Health, and Sociodemographic Characteristics across US Counties"

_ijerph, 2022, doi:10.3390/ijerph19116898_

Round 1

Reviewer 1 Report

The results of the paper are very intriguing and highly interesting. The obtained  "hockey stick" shape of the piecewise linear dependency of the health score on the environmental amenities is an empirical result which loudly calls for some theoretical explanation. Although I am rather sceptical about interpretations given by the Authors (also, there are not enogh of them) but even the empirical result by itself is worth publishing.  

However, I have some comments and suggestion to improve the paper. 

First of all, more details on raw (or semi-raw) empirical data are needed. I assume that you have a lot of data (health scores and NAS value for every participant?) but it would be good to show as much of them as possible. For instant, to represent them on the plane (the same as on Fig. 1 and Fig. 2) as points or, even better, as aggregated points (e.g., to divide the plane on small, but not too small, squares and show the number of cases in every square), or to give average values for any considered county.    

Line 17:  "there was no relationship in low amenities areas"   it is not clear what "relationship" is here referred to by the Authors.

The Authors should briefly presents the statistical methods used (I mean the end of section 2), especially the AIC values, but few words about this particular implementation of LOWESS could be relevant, as well. In particular, please provide more reader-friendly description of results from Table 2 and Table 3.

The results of this paper seem to be really interesting, potentially of interest for a broad audience, and it would be good to make them more accessible for a wider public (also for those withot working knowlege of these statistical and mathematical tools). 

I also see an important variable always associated with the environmental issues: the rural/urban place of residence of participants of the survey. It seems that this variable was not taken into account.  In my opinion, this is very essential whether a person lives in a village (small towns could be included in this category as well) or in metropolitan areas.  Can you say anythig about that issue? Otherwise, the study has a kind of  methodological problem unresolved.

My main question concerning the interpretaion of the empirical results is: why the inflection point on all curves is  NAS=0.  Although it seems natural, that the curves should be rather flat on the right end, and of high slope on the left end, but why the curve slope changes only once, only at NAS=0, and why the change is  so abrupt? Please either try to explain this phenomenon, or explicitly state it as an important open problem for further research.

My overall recommendation is a major revision along lines indicated above.

Author Response

Reviewer #1

The results of the paper are very intriguing and highly interesting. The obtained "hockey stick" shape of the piecewise linear dependency of the health score on the environmental amenities is an empirical result which loudly calls for some theoretical explanation. Although I am rather sceptical about interpretations given by the Authors (also, there are not enogh of them) but even the empirical result by itself is worth publishing.  However, I have some comments and suggestion to improve the paper. 

Thank you for the review. Your feedback has made this a stronger paper!

First of all, more details on raw (or semi-raw) empirical data are needed. I assume that you have a lot of data (health scores and NAS value for every participant?) but it would be good to show as much of them as possible. For instant, to represent them on the plane (the same as on Fig. 1 and Fig. 2) as points or, even better, as aggregated points (e.g., to divide the plane on small, but not too small, squares and show the number of cases in every square), or to give average values for any considered county.    

We add 2 new figures that include the scatter plot points of the individual participant scores. Because these values add clutter to the graphic, we decided to create these as supplemental graphics. Please see supplemental figures 1 & 2.

Line 17:  "there was no relationship in low amenities areas"   it is not clear what "relationship" is here referred to by the Authors.

The abstract has been updated to make it more clear.  

The Authors should briefly presents the statistical methods used (I mean the end of section 2), especially the AIC values, but few words about this particular implementation of LOWESS could be relevant, as well.

We added in AIC values in the first paragraph of page 7.  We also added in a description of the LOWESS function in the statistical analysis section on page 4.

In particular, please provide more reader-friendly description of results from Table 2 and Table 3.The results of this paper seem to be really interesting, potentially of interest for a broad audience, and it would be good to make them more accessible for a wider public (also for those withot working knowlege of these statistical and mathematical tools). 

We enhanced the footnote of table 2 to enhance readability and interpretability and to make it more accessible.

I also see an important variable always associated with the environmental issues: the rural/urban place of residence of participants of the survey. It seems that this variable was not taken into account.  In my opinion, this is very essential whether a person lives in a village (small towns could be included in this category as well) or in metropolitan areas.  Can you say anythig about that issue? Otherwise, the study has a kind of  methodological problem unresolved.

Thank you for the helpful comment. We agree with the reviewer and have added urban-rural status as another subgroup to the analysis. Although including this as a covariate in the model did not change the multivariable models, there were important subgroup differences. Please see the updated tables 2 and 3.  

My main question concerning the interpretaion of the empirical results is: why the inflection point on all curves is  NAS=0.  Although it seems natural, that the curves should be rather flat on the right end, and of high slope on the left end, but why the curve slope changes only once, only at NAS=0, and why the change is  so abrupt? Please either try to explain this phenomenon, or explicitly state it as an important open problem for further research.

We have added a sentence in the limitations explicitly stating that this is an open problem and needs more research.

Reviewer 2 Report

The study explored relationships among the characteristics of natural environment, health, and sociodemographic characteristics across 119 counties in the U.S. The authors used large scale data about participants health and associated them with characteristics of environments, where they are living. From this point, the research is novel and important.

The physical characteristics of environments were described by USDA Economic search Service’s Natural Amenities Scale. The scale tried to describe qualities of the places that enhance the location as a place to live. However, I did not find on the WWW pages of the USDA Economic search Service’s any theory that is behind the construction of the scale. For instance, why warm winter would enhance the location as a place to live? I think that the construction of this scale needs some explanation and maybe, also mention its limitations.

Second point is the sample. If I understand well, as the sample people with chronic conditions from 44 primary care practices served. If so, there is problem that the study did not say anything about health of general population in specific areas, but it compares only samples of people with chronic health diseases in the specific areas.  Is the idea of the study based on an assumption than in healthier places people with chronic diseases are healthier than people with chronic diseases in less healthy places? This is an important drawback of the study, therefore it should be explicitly stated in the limitation of the study and explained and justified, why comparing these samples reflects health condition of the population in the specific areas.  

It should be explained, why the authors divided sample for race (white vs. other) and ethnicity (Hispanic vs. Non- Hispanic). Readers outside the U.S. may not understand to these differences (race x ethnicity).

In the discussion, I recommend authors also consider findings from geographical psychology studies conducted in U.S. These authors refer differences in personalities among U.S. states, which might be also a factor influencing behavior and more/less healthier style of life.

Here are the studies:

Rentfrow, P. J., Jost, J. T., Gosling, S. D., & Potter, J. (2009). Statewide differences in personality predict voting patterns in 1996–2004 US presidential elections. Social and psychological bases of ideology and system justification, 1, 314-349.

Rentfrow, P. J., Gosling, S. D., Jokela, M., Stillwell, D. J., Kosinski, M., & Potter, J. (2013). Divided we stand: three psychological regions of the United States and their political, economic, social, and health correlates. Journal of personality and social psychology, 105(6), 996.

Rentfrow, P. J., & Jokela, M. (2016). Geographical psychology: The spatial organization of psychological phenomena. Current Directions in Psychological Science, 25(6), 393-398.

In addition, it should be also taken in an account a cultural specificity of the finding. Maybe, that it could be also indicated in the title of the study (e.g., “Nonlinear relationships among the natural environment, health, and sociodemographic characteristics across US counties”).

The cultural specificity also means that there are countries (such as Finland, Norway, etc.), where low temperature in winter did not discourage people to move outside and perform various winter sports.  

Author Response

Reviewer 2

The study explored relationships among the characteristics of natural environment, health, and sociodemographic characteristics across 119 counties in the U.S. The authors used large scale data about participants health and associated them with characteristics of environments, where they are living. From this point, the research is novel and important.

Thank you for the review. Your feedback has made this a stronger paper!

The physical characteristics of environments were described by USDA Economic search Service’s Natural Amenities Scale. The scale tried to describe qualities of the places that enhance the location as a place to live. However, I did not find on the WWW pages of the USDA Economic search Service’s any theory that is behind the construction of the scale. For instance, why warm winter would enhance the location as a place to live? I think that the construction of this scale needs some explanation and maybe, also mention its limitations.

Thank you for the review. Your feedback has made this a stronger paper!

Second point is the sample. If I understand well, as the sample people with chronic conditions from 44 primary care practices served. If so, there is problem that the study did not say anything about health of general population in specific areas, but it compares only samples of people with chronic health diseases in the specific areas.  Is the idea of the study based on an assumption than in healthier places people with chronic diseases are healthier than people with chronic diseases in less healthy places? This is an important drawback of the study, therefore it should be explicitly stated in the limitation of the study and explained and justified, why comparing these samples reflects health condition of the population in the specific areas.  

Yes, the idea was based on the assumption that in healthier places people with chronic diseases are healthier than people with chronic diseases in less healthy places. We only had information on people with chronic conditions. This is an important limitation to the study and limits generalizability. We have added to the limitations section.

It should be explained, why the authors divided sample for race (white vs. other) and ethnicity (Hispanic vs. Non- Hispanic). Readers outside the U.S. may not understand to these differences (race x ethnicity).

Race and ethnicity are considered different constructs in the US to allow for the classification of individuals within any race and simultaneously as Hispanic or non-Hispanic cultural groups.

The majority of our sample were non-Hispanic (92%) and white (80%). Small sample sizes limited further categorization of race.

In the discussion, I recommend authors also consider findings from geographical psychology studies conducted in U.S. These authors refer differences in personalities among U.S. states, which might be also a factor influencing behavior and more/less healthier style of life. Here are the studies:

Rentfrow, P. J., Jost, J. T., Gosling, S. D., & Potter, J. (2009). Statewide differences in personality predict voting patterns in 1996–2004 US presidential elections. Social and psychological bases of ideology and system justification, 1, 314-349.

Rentfrow, P. J., Gosling, S. D., Jokela, M., Stillwell, D. J., Kosinski, M., & Potter, J. (2013). Divided we stand: three psychological regions of the United States and their political, economic, social, and health correlates. Journal of personality and social psychology, 105(6), 996.

Rentfrow, P. J., & Jokela, M. (2016). Geographical psychology: The spatial organization of psychological phenomena. Current Directions in Psychological Science, 25(6), 393-398.

These literature

In addition, it should be also taken in an account a cultural specificity of the finding. Maybe, that it could be also indicated in the title of the study (e.g., “Nonlinear relationships among the natural environment, health, and sociodemographic characteristics across US counties”).

Thank you. We have updated the title based on the reviewers suggestion.

The cultural specificity also means that there are countries (such as Finland, Norway, etc.), where low temperature in winter did not discourage people to move outside and perform various winter sports

We agree with the reviewer. There are some US states, similar to Finland and Norway, where low temperatures and snow promote winter sports. This is a limitation of the NAS.

Reviewer 3 Report

The paper presents a exploratory study on how the relationship between natural environments and health may exhibit non-linear associations. The authors used a cohort of individuals and county-level data to do this. They found a variety of results that indicate the complex phenomena of nature and health. As an exploratory study it points to future directions for research. Overall it is well written and nice study. A few items:

  • line 101 - put "See Figure 1" in parentheses. Same on line 122. And line 127. Review the paper for any other instances like this.
  • line 108 - is that p-value of < 0.2? Maybe be clearer that this was your cutoff for investigating these interactions further and not any sort of significance test.
  • Table 1 - says "Education, college degree or less" but earlier it is stated as "college graduate or more vs. associates degree or less" on line 88
  • Line 207 - "probably do not often differ within a county" sounds clunky. Could say "are likely not to differ drastically" or something  like that.
  • You could also mention other non-linear methods that could be explored in future research in your conclusion. Particularly I am thinking of general additive models (GAMs). Glad you did not just use exponentials to get the non-linear part.

One major thing is the comment about where 90% of the sample lived. It would be nice to see a sensitivity analysis with just this sample "the majority (90%) of participants reside in 24 counties" from line 214. Could you run your Stata code with just these people and see if a difference surfaced? I think that even just referencing this would better make the case for your results, that what you found wasn't majorly influenced by the smaller set.

Overall just some addressing format and this one modeling item and I think the paper will be nice. Depending on how easy it is to run your code on the subset this may be a minor revision.

Author Response

The paper presents a exploratory study on how the relationship between natural environments and health may exhibit non-linear associations. The authors used a cohort of individuals and county-level data to do this. They found a variety of results that indicate the complex phenomena of nature and health. As an exploratory study it points to future directions for research. Overall it is well written and nice study. A few items:

Thank you for the review. Your feedback has made this a stronger paper!

·         line 101 - put "See Figure 1" in parentheses. Same on line 122. And line 127. Review the paper for any other instances like this.

These have been updated. We also did the same for the three “See Table #” statements.

·         line 108 - is that p-value of < 0.2? Maybe be clearer that this was your cutoff for investigating these interactions further and not any sort of significance test.

This sentence has been updated. Thank you for the recommendation.

·         Table 1 - says "Education, college degree or less" but earlier it is stated as "college graduate or more vs. associates degree or less" on line 88

Good catch, thank you. College graduate or more, is correct.

·         Line 207 - "probably do not often differ within a county" sounds clunky. Could say "are likely not to differ drastically" or something  like that.

The language has been updated.

·         You could also mention other non-linear methods that could be explored in future research in your conclusion. Particularly I am thinking of general additive models (GAMs). Glad you did not just use exponentials to get the non-linear part.

This has been added. Thank you.

One major thing is the comment about where 90% of the sample lived. It would be nice to see a sensitivity analysis with just this sample "the majority (90%) of participants reside in 24 counties" from line 214. Could you run your Stata code with just these people and see if a difference surfaced? I think that even just referencing this would better make the case for your results, that what you found wasn't majorly influenced by the smaller set.

Thank you for the thoughtful comment. We have performed a sensitivity analysis and added it to the results section.  There were no major differences noted.

Overall just some addressing format and this one modeling item and I think the paper will be nice. Depending on how easy it is to run your code on the subset this may be a minor revision.

Round 2

Reviewer 1 Report

The revision is satisfactory. The answers for my comments is well written and address all my queries. I wish good luck to the Authors in further studies 

Reviewer 2 Report

I thank authors to react on my suggestions and reccomendations.

Reviewer 3 Report

Good job addressing the comments from reviewers. I think that paper is improved and provides a good background for further investigation.